# Mapping cognitive ontologies to and from the brain

**Yannick Schwartz, Bertrand Thirion, and Gael Varoquaux**
Parietal Team, Inria Saclay Ile-de-France
Saclay, France
`firstname.lastname@inria.fr`

## Abstract

Imaging neuroscience links brain activation maps to behavior and cognition via correlational studies. Due to the nature of the individual experiments, based on eliciting neural response from a small number of stimuli, this link is incomplete, and unidirectional from the causal point of view. To come to conclusions on the function implied by the activation of brain regions, it is necessary to combine a wide exploration of the various brain functions and some inversion of the statistical inference. Here we introduce a methodology for accumulating knowledge towards a bidirectional link between observed brain activity and the corresponding function. We rely on a large corpus of imaging studies and a predictive engine. Technically, the challenges are to find commonality between the studies without denaturing the richness of the corpus. The key elements that we contribute are labeling the tasks performed with a cognitive ontology, and modeling the long tail of rare paradigms in the corpus. To our knowledge, our approach is the first demonstration of predicting the cognitive content of completely new brain images. To that end, we propose a method that predicts the experimental paradigms across different studies.

## 1  Introduction

Functional brain imaging, in particular fMRI, is the workhorse of brain mapping, the systematic study of which areas of the brain are recruited during various experiments. To date, 33K papers on pubmed mention "fMRI", revealing an accumulation of activation maps related to specific tasks or cognitive concepts. From this literature has emerged the notion of brain modules specialized to a task, such as the celebrated fusiform face area (FFA) dedicated to face recognition [1]. However, the link between the brain images and high-level notions from psychology is mostly done manually, due to the lack of co-analysis framework. The challenges in quantifying observations across experiments, let alone at the level of the literature, leads to incomplete pictures and well-known fallacies. For instance a common trap is that of *reverse inferences* [2]: attributing a cognitive process to a brain region, while the individual experiments can only come to the conclusion that it is recruited by the process under study, and not that the observed activation of the region demonstrates the engagement of the cognitive process. Functional specificity can indeed only be measured by probing a large variety of functions, which exceeds the scale of a single study. Beyond this lack of specificity, individual studies are seldom comprehensive, in the sense that they do not recruit every brain region.

Prior work on such large scale cognitive mapping of the brain has mostly relied on coordinate-based meta-analysis, that forgo activation maps and pool results across publications via the reported Talairach coordinates of activation foci [3, 4]. While the underlying thresholding of statistical maps and extraction of local maxima leads to a substantial loss of information, the value of this approach lies in the large amount of studies covered: Brainmap [3], that relies on manual analysis of the literature, comprises 2 298 papers, while Neurosynth [4], that uses text mining, comprises 4 393 papers. Such large corpuses can be used to evaluate the occurrence of the cognitive and behavioral

terms associated with activations and formulate reverse inference as a Bayesian inversion on standard (forward) fMRI inference [2, 4]. On the opposite end of the spectrum, [5] shows that using a machine-learning approach on studies with different cognitive content can predict this content from the images, thus demonstrating principled reverse inference across studies. Similarly, [6] have used image-based classification to challenge the vision that the FFA is by itself specific of faces. Two trends thus appear in the quest for explicit correspondences between brain regions and cognitive concepts. One is grounded on counting term frequency on a large corpus of studies described by coordinates. The other uses predictive models on images. The first approach can better define functional specificity by avoiding the sampling bias inherent to small groups of studies; however each study in a coordinate-based meta-analysis brings only very limited spatial information [7].

Our purpose here is to outline a strategy to accumulate knowledge from a brain functional image database in order to provide grounds for principled bidirectional reasoning from brain activation to behavior and cognition. To increase the breadth in co-analysis and scale up from [5], which used only 8 studies with 22 different cognitive concepts, we have to tackle several challenges. A first challenge is to find commonalities across studies, without which we face the risk of learning idiosyncrasies of the protocols. For this very reason we choose to describe studies with terms that come from a cognitive paradigm ontology instead of a high-level cognitive process one. This setting enables not only to span the terms across all the studies, but also to use atypical studies that do not clearly share cognitive processes. A second challenge is that of diminishing statistical power with increasing number of cognitive terms under study. Finally, a central goal is to ensure some sort of functional specificity, which is hindered by the data scarcity and ensuing biases in an image database.

In this paper, we gather 19 studies, comprising 131 different conditions, which we labeled with 19 different terms describing experimental paradigms. We perform a brain mapping experiment across these studies, in which we consider both forward and reverse inference. Our contributions are two-fold: on the one hand we show empirical results that outline specific difficulties of such co-analysis, on the second hand we introduce a methodology using image-based classification and a cognitive-paradigm ontology that can scale to large set of studies. The paper is organized as following. In section 2, we introduce our methodology for establishing correspondence between studies and performing forward and reverse inference across them. In section 3, we present our data, a corpus of studies and the corresponding paradigm descriptions. In section 4 we show empirically that our approach can predict these descriptions in unseen studies, and that it gives promising maps for brain mapping. Finally, in section 5, we discuss the empirical findings in the wider context of meta-analyses.

## 2  Methodology: annotations, statistics and learning

### 2.1  Labeling activation maps with common terms across studies

A standard task-based fMRI study results in *activation maps* per subject that capture the brain response to each experimental condition. They are combined to single out responses to high-level cognitive functions in so-called *contrast maps*, for which the inference is most often performed at the group level, across subjects. These contrasts can oppose different experimental conditions, some to capture the effect of interest while others serve to cancel out non-specific effects. For example, to highlight computation processes, one might contrast *visual calculation* with *visual sentences*, to suppress the effect of the stimulus modality (visual instructions), and the explicit stimulus (reading the numbers).

When considering a corpus of different studies, finding correspondences between the effects highlighted by the contrasts can be challenging. Indeed, beyond classical *localizers*, capturing only very wide cognitive domains, each study tends to investigate fairly unique questions, such as syntactic structure in language rather than language in general [8]. Combining the studies requires engineering *meta-contrasts* across studies. For this purpose, we choose to affect a set of *terms* describing the content of each condition. Indeed, there are important ongoing efforts in cognitive science and neuroscience to organize the scientific concepts into formal ontologies [9]. Taking the ground-level objects of these gives a suitable family of terms, a *taxonomy* to describe the experiments.

## 2.2 Forward inference: which regions are recruited by tasks containing a given term?

Armed with the term labels, we can use the standard fMRI analysis framework and ask using a General Linear Model (GLM) across studies for each voxels of the subject-level activation images if it is significantly-related to a term in the corpus of images. If $\boldsymbol{x} \in \mathbb{R}^p$ is the observed activation map with $p$ voxels, the GLM tests $\mathcal{P}(\boldsymbol{x}_i \neq 0|T)$ for each voxel $i$ and term $T$. This test relies on a linear model that assumes that the response in each voxel is a combination of the different factors and on classical statistics:

$$\boldsymbol{x} = \boldsymbol{Y}\boldsymbol{\beta} + \varepsilon,$$

where $\boldsymbol{Y}$ is the design matrix yielding the occurrence of terms and $\boldsymbol{\beta}$ the term effects. Here, we assemble *term-versus-rest* contrasts, that test for the specific effect of the term. The benefit of the GLM formulation is that it estimates the effect of each term partialing out the effects of the other terms, and thus imposes some form of functional specificity in the results. Term co-occurrence in the corpus can however lead to collinearity of the regressors.

## 2.3 Reverse inference: which regions are predictive of tasks containing a given term?

Poldrack 2006 [2] formulates reverse inferences as reasoning on $\mathcal{P}(T|\boldsymbol{x})$, the probability of a term $T$ being involved in the experiment given the activation map $\boldsymbol{x}$. For coordinate-based meta analysis, as all that is available is the presence or the absence of significant activations at a given position, the information on $\boldsymbol{x}$ boils down to $\{i, \boldsymbol{x}_i \neq 0\}$. Approaches to build a reverse inference framework upon this description have relied on Bayesian inversion to go from $\mathcal{P}(\boldsymbol{x}_i \neq 0|T)$, as output by the GLM, to $\mathcal{P}(T|\boldsymbol{x}_i \neq 0)$ [2, 4]. In terms of predictive models on images, this approach can be understood as a naive Bayes predictor: the distribution of the different voxels are learned independently conditional to each term, and Bayes' rule is used for prediction. Learning voxels-level parameters independently is a limitation as it makes it harder to capture distributed effects, such as large-scale functional networks, that can be better predictors of stimuli class than localized regions [6]. However, learning the full distribution of $\boldsymbol{x}$ is ill-posed, as $\boldsymbol{x}$ is high-dimensional. For this reason, we must resort to statistical learning tools.

We choose to use an $\ell_2$-regularized logistic regression to directly estimate the conditional probability $\mathcal{P}(T|\boldsymbol{x})$ under a linear model. The choice of linear models is crucial to our brain-mapping goals, as their decision frontier is fully represented by a brain map[1] $\boldsymbol{\beta} \in \mathbb{R}^p$. However, as the images are spatially smooth, neighboring voxels carry similar information, and we use feature clustering with spatially-constrained Ward clustering [10] to reduce the dimensionality of the problem. We further reduce the dimensionality by selecting the most significant features with a one-way ANOVA. We observe that the classification performance is not hindered if we reduce the data from 48K voxels to 15K parcels[2] and then select the 30% most significant features. The classifier is quite robust to these parameters, and our choice is motivated by computational concerns. We indeed use a leave-one-study out cross validation scheme, nested with a 10-fold stratified shuffle split to set the $\ell_2$ regularization parameter. As a result, we need to estimate 1200 models per term label, which amounts to over 20K in total. The dimension reduction helps making the approach computationally tractable.

The learning task is rendered difficult by the fact that it is highly multi-class, with a small number of samples in some classes. To divide the problem in simpler learning tasks, we use the fact that our terms are derived from an ontology, and thus can be grouped by parent category. In each category, we apply a strategy similar to one-versus-all: we train a classifier to predict the presence of each term, opposed to the others. The benefits of this approach are *i)* that it is suited to the presence of multiple terms for a map, and *ii)* that the features it highlights are indeed selective for the associated term only.

Finally, an additional challenge faced by the predictive learning task is that of strongly imbalanced classes: some terms are very frequent, while others hardly present. In such a situation, an empirical risk minimizer will mostly model the majority class. Thus we add sample weights inverse of the

| CATEGORY | TERMS |
|---|---|
| Stimulus modality | visual, auditory |
| Explicit stimulus | words, shapes, digits, abstract patterns, non-vocal sounds, scramble, face |
| Instructions | attend, read, move, track, count, discriminate, inhibit |
| Overt response | saccades, none, button press |

Table 1: Subset of CogPO terms and categories that are present in our corpus

population imbalance in the training set. This strategy is commonly used to compensate for covariate shift [11]. However, as our test set is drawn from the same corpus, and thus shows the same imbalance, we apply an inverse bias in the decision rule of the classifier by shifting the probability output by the logistic model: if $P$ is the probability of the term presence predicted by the logistic, we use: $P_{\text{biased}} = \rho_{\text{term}} P$, where $\rho_{\text{term}}$ is the fraction of train samples containing the term.

# 3 An image database

## 3.1 Studies

We need a large collection of task fMRI datasets to cover the cognitive space. We also want to avoid particular biases regarding imaging methods or scanners, and therefore prefer images from different teams. We use 19 studies, mainly drawn from the OpenfMRI project [12], which despite remaining small in comparison to coordinate databases, is as of now the largest open database for task fMRI. The datasets include risk-taking tasks [13, 14], classification tasks [15, 16, 17], language tasks [18, 8, 19], stop-signal tasks [20], cueing tasks [21], object recognition tasks [22, 23], functional localizers tasks [24, 25], and finally a saccades & arithmetic task [26]. The database accounts for 486 subjects, 131 activation map types, and 3 826 individual maps, the number of subjects and map types varying across the studies. To avoid biases due to heterogeneous data analysis procedures, we re-process from scratch all the studies with the SPM (Statistical Parametric Mapping) software.

## 3.2 Annotating

To tackle highly-multiclass problems, computer vision greatly benefits from the WordNet ontology [27] to standardize annotation of pictures, but also to impose structure on the classes. The neuroscience community recognizes the value of such vocabularies and develops ontologies to cover the different aspects of the field such as protocols, paradigms, brain regions and cognitive processes. Among the many initiatives, CogPO (The Cognitive Paradigm Ontology) [9] aims to represent the cognitive paradigms used in fMRI studies. CogPO focuses on the description of the experimental conditions characteristics, namely the explicit stimuli and their modality, the instructions, and the explicit responses and their modality. Each of those categories use standard terms to specify the experimental condition. As an example a stimulus modality may be *auditory* or *visual*, the explicit stimulus a *non-vocal sound* or a *shape*. We use this ontology to label with the appropriate terms all the experimental conditions from the database. The categories and terms that we use are listed in Table 1.

# 4 Experimental results

## 4.1 Forward inference

In our corpus, the occurrence of some terms is too correlated and gives rise to co-linear regressors. For instance, we only have visual or auditory stimulus modalities. While a handful of contrasts display both stimulus modalities, the fact that a stimulus is not auditory mostly amounts to it being visual. For this reason, we exclude from our forward inference *visual*, which will be captured by negative effects on *auditory*, and *digits*, that amounts mainly to the instruction being *count*. We fit the GLM using a design matrix comprising all the remaining terms, and consider results with p-values corrected for multiple comparisons at a 5% family-wise error rate (FWER). To evaluate the spatial layout of the different CogPO categories, we report the different term effects as outlines in the brain, and show the 5% top values for each term to avoid clutter in Figure 3. Forward inference

outlines many regions relevant to the terms, such as the primary visual and auditory systems on the *stimulus modality* maps, or pattern and object-recognition areas in the ventral stream, on the *explicit stimulus* maps.

It can be difficult to impose a functional specificity in forward inference because of several phenomena: *i)* the correlation present in the design matrix, makes it hard to separate highly associated (often anti-correlated) factors, as can be seen in Fig. 1, right. *ii)* the assumption inherent to this model that a certain factor is expressed identically across all experiments where it is present. This assumption ignores modulations and interactions effects that are very likely to occur; however their joint occurrence is related to the protocol, making it impossible to disentangle these factors with the database used here. *iii)* important confounding effects are not modeled, such as the effect of attention. Indeed the *count* map captures networks related to visuo-spatial orientation and attention: a dorsal attentional network, and a salience network (insulo-cingulate network [28]) in Figure 3.

## 4.2   Reverse inference

The promise of predictive modeling on a large statistical map database is to provide principled reverse inference, going from observations of neural activity to well-defined cognitive processes. The classification model however requires a careful setting to be specific to the intended effect. Figure 1 highlights some confounding effects that can captured by a predictive model: two statistical maps originating from the same study are closer than two maps labeled as sharing a same experimental condition in the sense of a Euclidean distance. We mitigate the capture of undesired effect with different strategies. First we use term labels at span across studies, and refrain from using those that were not present in at least two. We ensure this way that no term is attached to a specific study. Second, we only test the classifiers on previously unseen studies and if possible subjects, using for example a leave-one-study out cross validation scheme. A careless classification setting can very easily lead to training a study detector.

Figure 2 summarizes the highly multi-class and imbalanced problem that we face: the distribution of the number of samples per class displays a long tail. To find non-trivial effects we need to be able to detect the under-represented terms as well as possible. As a reference method, we use a K-NN, as it is in general a fairly good approach for highly multi-class problems. Its training is independent of the term label structure and predicts the map labels instead. It subsequently assigns to a new map terms that are present in more than half of its nearest neighbors from the training[3]. We compare this approach to training independent predictive models for each term and use three types of classifiers: a naive Bayes, a logistic regression, and a weighted logistic regression. Figure 2 shows the results for each method in terms of precision and recall, standard information-retrieval metrics. Note that the performance scores mainly follow the class representation, *i.e.* the number of samples per class in the train set. Considering that rare occurrences are also those that are most likely to provide new insight, we want a model that promotes recall over precision in the tail of the term frequency distribution. On the other hand, well represented classes are easier to detect and correspond to massive, well-known mental processes. For these, we want to favor precision, *i.e.* not affecting the corresponding term to other processes, as these term are fairly general and non-descriptive.

Overall the K-NN has the worst performance, both in precision and recall. It confirms the idea outlined in Figure 1, that an Euclidean distance alone is not appropriate to discriminate underlying brain functions because of overwhelming confounding effects[4]. Similarly, the naive bayes performs poorly, with very high recall and low precisions scores which lead to a lack of functional specificity. On the contrary, the methods using a logistic regression show better results, and yield performance scores above the chance levels which are represented by the red horizontal bars for the leave-one-study out cross validation scheme in Figure 2. Interestingly, switching the cross validation scheme to a leave-one-laboratory out does not change the performance significantly. This result is important, as it confirms that the classifiers do not rely on specificities from the stimuli presentation in a research group to perform the prediction. We mainly use data drawn from 2 different groups in this work, and use those data in turn to train and test a logistic regression model. The predicitions scores for

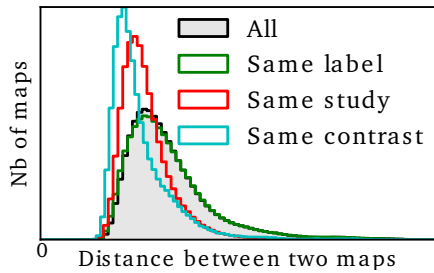
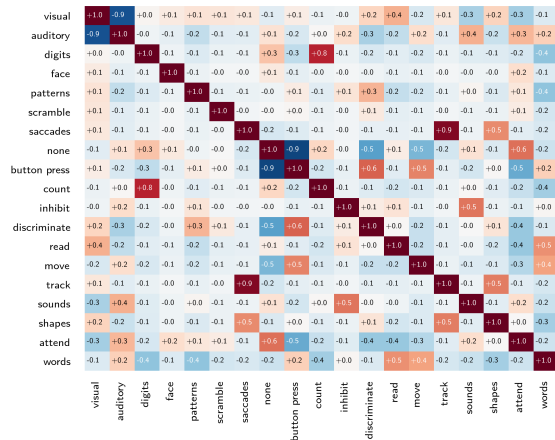

Figure 1: (Left) Histogram of the distance between maps owing to their commonalities: study of origin, functional labels, functional contrast. (Right) Correlation of the design matrix.

the terms present in both groups are displayed in Figure 2, with the chance levels represented by the green horizontal bars for this cross validation scheme.

We evaluate the spatial layout of maps representing CogPO categories for reverse inference as well, and report boundaries of the 5% top values from the weighted logistic coefficients. Figure 3 reports the outlined regions that include motor cortex activations in the *instructions* category, and activations in the auditory cortex and FFA respectively for the words and faces terms in the *explicit stimulus* category. Despite being very noisy, those regions report findings consistent with the literature and complementary to the forward inference maps. For instance, the *move* instruction map comprises the motor cortex, unlike for forward inference. Similarly, the *saccades* over response map segments the intra-parietal sulci and the frontal eye fields, which corresponds to the well known signature of saccades, unlike the corresponding forward inference map, which is very non specific of saccades[5].

## 5  Discussion and conclusion

Linking cognitive concepts to brain maps can give solid grounds to the diffuse knowledge derived in imaging neuroscience. Common studies provide evidence on which brain regions are recruited in given tasks. However coming to conclusions on the tasks in which regions are specialized requires data accumulation across studies to overcome the small coverage in cognitive domain of the tasks assessed in a single study. In practice, such a program faces a variety of roadblocks. Some are technical challenges, that of build a statistical predictive engine that can overcome the curse of dimensionality. While others are core to meta-analysis. Indeed, finding correspondence between studies is a key step to going beyond idiosyncrasies of the experimental designs. Yet the framework should not discard rare but repeatable features of the experiments as these provide richness to the description of brain function.

We rely on ontologies to solve the correspondence problem. It is an imperfect solution, as the labeling is bound to be inexact, but it brings the benefit of several layers of descriptions and thus enable us to fraction the multi-class learning task in simpler tasks. A similar strategy based on WordNet was essential to progress in object recognition in the field of computer vision [27]. Previous work [5] showed high classification scores for several mental states across multiple studies, using cross-validation with a leave-one-subject out strategy. However, as this work did not model common factors across studies, the mental state was confounded by the study. In every study, a subject was represented by a single statistical map, and there is therefore no way to validate whether the study or the mental state was actually predicted. As figure 1 shows, predicting studies is much easier albeit of little neuroscientific interest. Interestingly, [5] also explores the ability of a model to be predictive on two different studies sharing the same cognitive task, and a few subjects. When using the common subjects, their model performs worse than without these subjects, as it partially mistakes cognitive

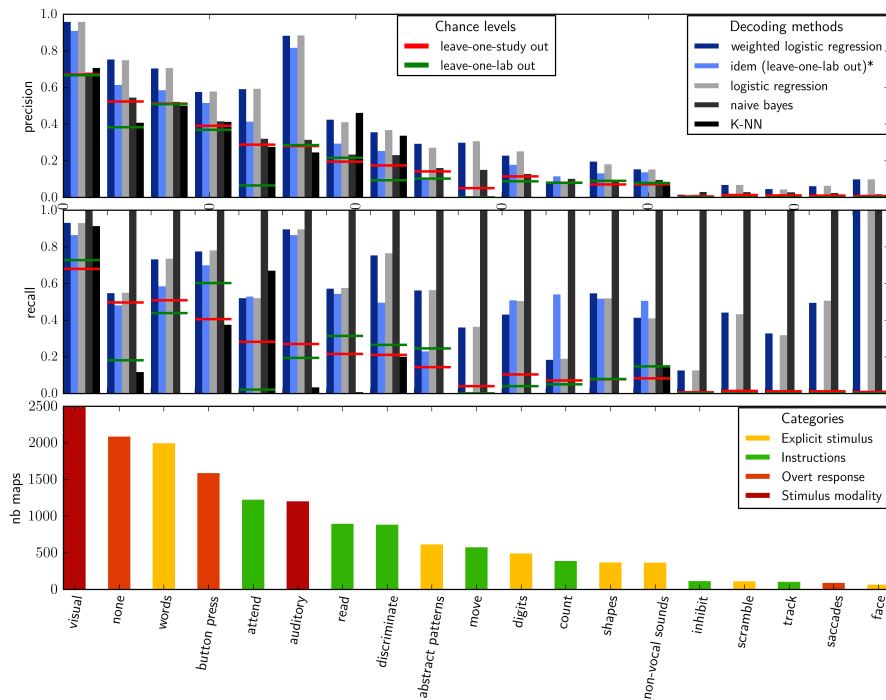

Figure 2: Precision and recall for all terms per classification method, and term representation in the database. The * denotes a leave-one-laboratory out cross validation scheme, associated with the green bars representing the chance levels. The other methods use a leave-one-study out cross validation, whose chance levels are represented by the red horizontal bars.

tasks for subjects. This performance drop illustrates that a classifier is not necessarily specific to the desired effect, and in this case detects subjects in place of tasks to a certain degree. To avoid this loophole, we included in our corpus only studies that had terms in common with at least on other study and performed cross-validation by leaving a study out, and thus predicting from completely new activation maps. The drawback is that it limits directly the number of terms that we can attempt to predict given a database, and explain why we have fewer terms than [5] although we have more than twice as many studies. Indeed, in [5], the terms cannot be disambiguated from the studies.

Our labeled corpus is riddled with very infrequent terms giving rise to class imbalance problems in which the rare occurrences are the most difficult to model. Interestingly, though coordinates databases such as Neurosynth [4] cover a larger set of studies and a broader range of cognitive processes, they suffer from a similar imbalance bias, which is given by the state of the literature. Indeed, by looking at the terms in Neurosynth, that are the closest to the one we use in this work, we find that *motor* is cited in 1090 papers, *auditory* 558, *word* 660, and the number goes as low as 55 and 31 for *saccade* and *calculation* respectively. Consequently, these databases may also yield inconsistent results. For instance, the reverse inference map corresponding to the term *digits* is empty, whereas the forward inference map is well defined [6]. Neurosynth draws from almost 5K studies while our work is based on 19 studies; however, unlike Neurosynth, we are able to benefit from the different contrasts and subjects in our studies, which provides us with 3 826 training samples. In this regard, our approach is particularly interesting and can hope to achieve competitive results with much less studies.

This paper shows the first demonstration of *zero-shot learning* for prediction of tasks from brain activity: paradigm description is given for images from unseen studies, acquired on different scanners, in different institutions, on different cognitive domains. More importantly than the prediction per se, we pose the foundation of a framework to integrate and co-analyze many studies. This data

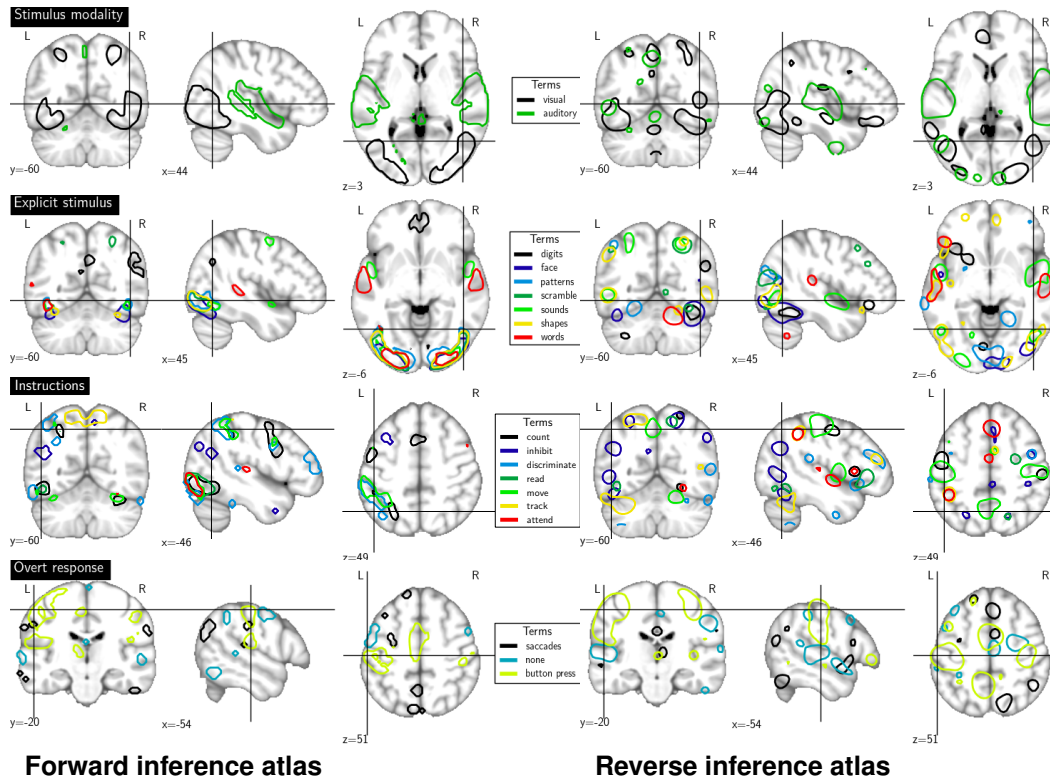

**Forward inference atlas**          **Reverse inference atlas**

Figure 3: Maps for the forward inference (left) and the reverse inference (right) for each term category. To minimize clutter, we set the outline so as to encompass 5% of the voxels in the brain on each figure, thus highlighting only the salient features of the maps. In reverse inference, to reduce the visual effect of the parcellation, maps were smoothed using a $\sigma$ of 2 voxels.

accumulation, combined with the predictive model can provide good proxies of *reverse inference maps*, giving regions whose activation supports certain cognitive functions. These maps should, in principle, be better suited for causal interpretation than maps estimated from standard brain mapping correlational analysis. In future work, we plan to control the significance of the reverse inference maps, that show promising results but would probably benefit from thresholding out non-significant regions. In addition, we hope that further progress, in terms of spatial and cognitive resolution in mapping the brain to cognitive ontologies, will come from enriching the database with new studies, that will bring more images, and new low and high-level concepts.

### Acknowledgments

This work was supported by the ANR grants BrainPedia ANR-10-JCJC 1408-01 and IRMGroup ANR-10-BLAN-0126-02, as well as the NSF grant NSF OCI-1131441 for the OpenfMRI project.

## Footnotes

[1]In this regard, the Naive Bayes prediction strategy does yield clear cut maps, as its decision boundary is a conic section.

[2]Reducing even further down to 2K parcels does not impact the classification performance, however the brain maps $\boldsymbol{\beta}$ are then less spatially resolved.

[3]K was chosen in a cross-validation loop, varying between 5 and 20. Such small numbers for K are useful to avoid penalizing under-represented terms of rare classes in the vote of the KNN. For this reason we do not explore above K=20, in respect to the small number of occurrences of the *faces* term.

[4]Note that the picture does not change when $\ell_1$ distances are used instead of $\ell_2$ distances.

[5]This failure of forward inference is probably due to the small sample size of saccades.

[6] http://neurosynth.org/terms/digits

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
