[Reviews · NeurIPS 2013]

Submitted by Assigned_Reviewer_3

The authors describe a new fMRI meta-analysis technique, which enables prediction of activity maps based on experimental condition (forward inference) AND prediction of experimental condition from activity maps (reverse inference). The paper is partially an empirical survey of the issues involved in meta-analysis (long-tail distribution of cognitive processes, within-study confounding effects) and a description of a specific approach for classifying brain images using a cognitive-paradigm ontology.

The paper is attacking an interesting problem, and pools data from 19 different studies (all of which they re-processed and standardized). Their reverse-inference approach makes sense, and they apply multiple types of classifiers in order to outline general trends in classification. They point out some surprising facts about meta-analysis; for example, activity maps are more similar within study (between conditions) than within condition (between studies), presumably due to particular imaging protocols or scanning hardware.

It doesn't seem that their approach really leverages the fact that their cognitive labels are in an ontology, except for the fact that their classifiers are trained one-versus-all within a parent category. They make an analogy with ImageNet (WordNet) but CogPO seems quite dissimilar; it has only two levels, siblings within a level are not necessarily disjoint (e.g. "scrambled" and "faces" can apply to the same stimulus), and it's not possible to train classifiers at multiple levels of generality as in ImageNet (e.g. a "dog" classifier can utilize data from many different dog classes).

There is not much technical depth to the paper - forward inference is accomplished with a standard GLM model, and reverse inference uses standard classifiers and feature selection. The statement that this is the "first demonstration of predicting the cognitive content of completely new brain images" is somewhat oversold, since they are essentially just doing cross-study decoding.

It is unclear whether the reverse inference maps they generate are scientifically useful - they are very noisy and make many counterintuitive predictions (e.g. voxels in visual regions predicting that the stimulus is audio-only, many voxels in cerebellum related to high-level tasks), and the authors do not investigate the issue of how to perform significance testing on these maps. The maps seem to have poor spatial resolution for known ventral temporal regions that should be associated with certain activations (i.e. FFA is poorly resolved, if at all, for face activations).

Response to Rebuttal:
-I agree that the task framing and the way in which examples are labeled are novel. My comment about novelty was stating that the prediction on new brain images is performed using logistic regression and existing feature selection methods, so there is isn't a substantial technical contribution in this section.
-The lack of validation for the reverse inference maps (which in my opinion are the most interesting and potentially impactful part of this paper) still leaves me with major concerns. Why are visual cortex voxels predicting that stimuli are non-visual? Why are some of the maps (e.g. the "explicit stimulus" maps in visual cortex) so asymmetrical between hemispheres? Why was no attempt made to compute significance thresholds, which will be essential for using this method in a scientific context? If 19 studies are too few to give reasonable results, what order of magnitude of studies is required?
Summary: This is an interesting and informative look at forward and reverse inference in meta-analysis, but doesn't have any substantial technical novelty. It is not clear whether the reverse inference maps generated by the method are of high enough quality to be a useful tool for neuroscientists.

Submitted by Assigned_Reviewer_5

The authors provide a methodology for analyzing fMRI data from a database containing 19 studies, which allows making reverse inferences of brain regions specifically related to a given term, as opposed to forward inferences that are usually used in fMRI studies but that do not provide specificity. The authors apply this method to predict cognitive ontology terms for new brain images (not used in training). Differently from ref.[28] (Yarkoni et al., 2011), the authors do not use peak coordinates text-mined from journal articles, but analyze actual fMRI contrasts annotated with specific terms.

Multivoxel pattern recognition studies and automated meta-analyses over numerous neuroimaging experiments are becoming increasingly popular and it's only a question of time when they will become a common practice in the neuroimaging field, which still suffers from problems of low statistical power, simplistic assumptions (of candidate regions) and limited comparability. The current paper provides an interesting methodology for obtaining reasonably accurate reverse inferences using a relatively small number of studies with few subjects. Although a similar database has been used for classification of mental states in ref.[18] (Poldrack et al., 2009), the current paper focuses on several difficult issues not resolved in [18], such as predictability across different studies, mitigating various confounding effects, and using cognitive ontology terms instead of classes based on task/study.

Although the paper is good and appropriate for NIPS, it could be improved in several ways:
- the last sentence of the abstract is too general and a more specific claim of novelty (such as predicting ontology terms across different studies) should be used;
- to my knowledge, ref.[28] does not only use Talairach coordinates, as stated in the beginning of para.2 of the introduction;
- assertion in section 3.1 that images from many different teams are used in the database does not seem to be very accurate, as the vast majority of studies seem to have only 2 different senior authors (one of which is sometimes the first author). More specific information regarding the number of different scanners and truly independent study designs should be reported. It should also be discussed whether prediction across different studies is successful also for these truly independent studies (i.e. with no related study being in the training set) or only for similar studies (when a training set contains at least one study with shared authors, scanners or designs). Currently only mean performances are reported, which are not very informative in this regard.
- It would be useful to discuss why "buttons" and not "saccades" are inversely correlated with "none" (section 4.1);
- measures of "precision" and "recall" should be explicitly defined in the text or footnotes even if they are standard in certain fields;
- what are the chance levels for predictions in fig.2? - without them it's very difficult (or impossible) to evaluate the quality of predictions.
- it is not clear what are the criteria of classification into 3 "operating regimes" in fig.2 and what exactly do these regimes mean (if nothing particular, such classification should be avoided);
- what is the rationale behind choices of coordinates for brain slices in fig.3 maps?

In addition, the following typos/errors should be corrected:
- in the abstract it should be "nature" instead of "natural" (line 2) and "bidirectional" better than "bilateral" (line 8);
- in the 3rd sentence of section 4.1 (lines 3-4) I suppose it should be "visual" instead of "not visual";
- in the 5th line from the end of page 7, it should be "albeit" instead of "ableit";
- on page 8, there should be "fewer" instead of "less" in line 5 of para.1 and the last line of para.2, "many" instead of "much" in the last line of para.1 and "whereas" instead of "will" in line 9 of para.2.

Finally, it's more conventional for references to be numbered in the order of their presentation, not alphabetically.

---

The authors addressed most of my major concerns regarding weaknesses of the study, therefore, assuming their responses and additional data are incorporated into the final version and that minor points are corrected as well, this is a solid paper that should be useful for the NIPS community. A few more points regarding author responses:
- The analysis of training on Dehaene's data / testing on Poldrack's and vice versa is interesting and should be mentioned in the final version as well. It is not clear which term is "read" (no such term appeared in the paper) More detailed explanations regarding the substantial deterioration of "auditory" scores would be helpful.
- Chance levels should also be provided for the Poldrack vs. Dehaene testing. It is also a bit strange how the "saccades" term with both precision and recall close to 0 (fig. 2) can be above chance.
- I agree with Reviewer 3 that fig. 3 should be presented/explained in a more informative way, indicating statistical significances. As the involvement of cerebellum in cognition is not so well known as in motor response, this result is particularly interesting and it would be useful to discuss it in the text. Regarding the prediction of auditory stimuli by voxels in the visual cortex, it's not clear based on the shown data if the result is genuine, but if it is, it may be related to evidence of multisensory integration between primary sensory areas (see the work of Micah M. Murray)
Summary: The authors present a methodology of accumulating fMRI data from different studies and contrasts marked by different terms of cognitive ontology and use it for prediction of these terms based on new brain images. Although the paper can be improved in certain aspects, it presents interesting and important advances in neuroinformatics of functional brain imaging.

Submitted by Assigned_Reviewer_6

The authors present a novel method to link psychological concepts to fMRI activations using an automated scheme. In contradistinction to coordinate-based meta analyses, the authors rely on whole-brain activation maps from a free database. Building on previous research that used 8 studies in a meta-analysis, the authors extend the scope to 19 studies. Using the cognitive paradigm ontology to describe studies, the authors build a model that allows forward (given a word in a task, which brain regions activate) and reverse inference (given an activated brain region, which words did the task consist of).

Positively, the methods of the manuscript are sound, and the way methodological challenges are met is creative. The manuscript is clearly and well written, and demonstrates the challenges in application of the described approach to a slightly larger than before number of studies.

On the negative side, I think the authors should make clearer how their approach can scale to a larger number of studies. The number of evaluated studies is only incremental compared to previous research (~twice as much). So how can the approach be extended to e.g. 100 or 1000 studies? The authors repeatedly state that manual intervention (manual crafting) in the process is required. How will this manual crafting work for more studies?

In a similar vein: The empirical results presented look very reasonable, but not very surprising. Given common knowledge of cognitive neuroscience and upon reading the 19 studies meta-analysed by the authors, it seems the same results could be gained by the traditional, manual and qualitative way cognitive neuroscience works today. The authors must make a better point of how they think their model and quantification will go beyond this.
Summary: The authors describe a model quantitatively linking (forward and inverse reference) between cognitive concepts and brain activation. Although a step forward, it remains unclear how the approach can scale to higher number of studies.
Author Feedback

Author rebuttal: Reviewer 3 is concerned with the novelty of the contribution, and sees the methodology as "just doing cross-study decoding". The problem setting here is however very different to existing decoding studies, and completely novel: it performs annotation of images belonging to experimental conditions that are not represented in the training set. This is an important problem, since we cannot expect to ever have data for every possible experimental condition. Framing decoding as a multi-label learning task is new. The labels provide an intermediate representation which encodes the conditions, and makes it possible to annotate unknown conditions. This approach is related to "Zero-shot learning with semantic output codes" [Palatucci NIPS 2009], although that prior art did not address the multi-class challenge as it only demonstrated correct prediction of a word out of two. The novelty of our contribution also lies in its scale, as we not only reused the 8 studies from [18] [Poldrack, 2009] as reviewer 5 stated, but also extended the database to 19 studies.

Reviewer 5 raises an important issue regarding the true independence of the studies, as the datasets are mainly drawn from two research groups, and whether we could assess prediction performance taking this issue into consideration. We evaluate the precision performance by training the classifiers on data from one group, and on testing on the other one. Training on data from Dehaene's group (6 studies), we find 0.78 for visual, 0.34 for auditory, and 0.42 for read. Training on data from Poldrack's group (9 studies) we find 0.78 for visual, 0.67 for auditory, and 0.2 for read. These scores are comparable to those reported in the manuscript for the full dataset, 0.92 for visual, 0.79 for auditory, and 0.3 for read. The large difference in scores for auditive stimuli depending on the training group is due to the difference in class distribution (covariate shift). However, this clearly establishes that classification accuracy does not suffer from cross-lab data pooling.

Following reviewer 5's suggestion, we estimated the chance levels for the scores reported in figure 2. We find that only the "button" and "none" terms are not significantly above chance, and will update the manuscript with the chance levels for all terms.

Reviewer 6's main comment concerns the scale and scalability of our work. The scale itself is indeed larger than previous work only by a factor of 2. Previous approaches however become more challenging when they deal with more studies, as opposed to our approach that benefits from more data by breaking the multi-class problem in a multi-label problem. The difficulty in scaling up the annotation of images has been previously solved by manual work. For example, the brainmap.org database contains activation coordinates for over 10 thousand experiments, all annotated manually. Scaling up in a principled way is however a important challenge that is recognized and being addressed by the neuroinformatics community. Current projects develop automated techniques applying text mining to paper abstracts, in order to guide human experts in their annotation process. (J. Turner, 2013, "Automated Annotation of Abstracts for Cognitive Experiments", http://bio-ontologies.knowledgeblog.org/361?kblog-transclude=2).

We agree with reviewer 6 that our manuscript does not show neuroscientific results that could not be found with traditional studies. Our contribution is a methodology and a proof of concept that we believe can lead to new findings, but requires more datasets. This scarcity of data is actually linked to reviewers 3's comment on the maps (voxels in visual regions predicting for auditory stimuli), and reviewer 5's comment on the inverse correlation between overt responses "none" and "button". Those can be simply explained by the biases of our database. We only have auditory and visual stimuli: in other words predicting that a stimulus is not visual equals to predict the stimulus is auditory. Similarly we mainly have button responses, or no overt response, thus the inverse correlation. Reviewer 3 also notes that some voxels in the cerebellum are reported for high level tasks. This is not surprising at all, and well documented in the literature ([Schmahmann 2006], "Cognition, emotion and the cerebellum"). The benefit of our large-scale data-driven approach is that it highlights such findings, mitigating the bias of specific hypotheses, such as the study of cerebellum-based motor response. With regards to the quality of spatial maps, in particular spatial resolution, reverse inference is challenging and our results seem to us more convincing than prior art (see supplementary materials of [18] http://www.stanford.edu/group/memorylab/journalclubs/pdfs/Pol_PS09_supp.pdf).